# Perceptual learning mechanisms with single-stimulus exposure

**María del Carmen Sanjuán**⬚*◉, **James Byron Nelson**⬚◉

Procesos Psicológicos Básicos y su Desarrollo, University of the Basque Country (EHU), San Sebastian, País Vasco, Spain

◉ These authors contributed equally to this work.
* mariadelcarmen.sanjuan@ehu.eus

## Abstract

The present manuscript briefly reviews associative and non-associative theories of perceptual learning and presents two experiments (1a and 1b) examining the extent to which retrospective revaluation (RR) and a restoration of perceptual effectiveness (RPE) can account for the enhanced discriminability that experience with a single stimulus produces. In both experiments participants were exposed to compound visual stimuli BX, CY, and DZ in an online video-game method followed by conditioning with compound stimuli AX and AY. In Experiment 1a generalization to compounds BX, DX, BY, and DY was assessed. Generalization to compounds involving B, either BX or BY, was less than to DX or DY. The result possibly reflected enhanced salience of B, or that B had become inhibitory for the outcome through RR. Experiment 1b used a retardation test comparing compound stimuli BW and DW. Acquisition was more effective with BW than DW, suggesting that B had become more perceptually effective, resulting in it producing more external inhibition than D in Experiment 1a, and allowing it to condition more rapidly in Experiment 1b. Results are discussed with respect to current theories of perceptual learning and are consistent with a restoration of perceptual effectiveness.

## Introduction

Daily, people refine their ability to discriminate between similar stimuli even when direct experience is limited to a single exemplar. For example, we might recognize a new coffee as different from one previously tasted better than discriminating two new ones. This simple example illustrates that prior experience with a single stimulus can limit generalization to related stimuli. Such examples illustrate perceptual learning, investigated here, which are the processes involved in how the brain refines its ability to identify and distinguish between similar stimuli based on even limited exposure.

The present manuscript presents two experiments that examine mechanisms which may contribute to situations where experience with a single stimulus modifies

**Data availability statement:** The data are in the Open Science Foundation Repository in the "Perceptual learning mechanisms with single-stimulus exposure" project. The DOI linking to the project is DOI 10.17605/OSF.IO/TYMUB The link to the data is: https://osf.io/tymub/files/osfstorage/684af5c272c47fcffaa4d9a3.

**Funding:** M.C. S and J.B.N. Research presented here was funded by grants PGC2018-097769-BC21 and PID2023-149399NB-C22 from Spanish Ministerio de Ciencia, Innovación y Universidades and IT-1501-22 from the Basque government. The funders did not play any role in the study, decision to publish or preparation of the manuscript.

**Competing interests:** No competing interest to declare.

the extent to which generalization between it and similar stimuli occur. Such an effect is important as it is not explicitly covered by most theories of perceptual learning, nor are the aspects of such learning necessarily expected by them.

Perceptual learning studies, with animals and humans (see [1] for review), have used stimuli that are made similar by including common elements in both (e.g., AX and BX). When one stimulus (AX) is later paired with an outcome, generalization to BX is typically observed which is assumed due to the shared elements X. Preexposure to AX and BX can reduce this generalization, with intermixed exposure (AX/BX/AX/BX/…) producing greater discrimination than blocked exposure (AX, AX, … BX, BX, …) e.g., [2–10]. Differences in generalization produced by intermixed and blocked exposure have been important in the development of perceptual learning theories.

An influential associative account driven by this intermixed-blocked effect of perceptual learning was proposed by McLaren, Kaye and Mackintosh and McLaren and Mackintosh [11,12] which formalizes many ideas found in the literature, e.g., [13–18]. They suggested that perceptual learning arises from inter-element associations. When a compound stimulus such as AX is presented, its elements (A and X) become bidirectionally linked(A↔X), allowing each to evoke the others internal representation, a process they termed unitization. Assuming internal activation is less salient than direct stimulation, repeated exposure reduces the overall salience of the compound and its components. This framework allows discrimination by way of two paths: differential losses of stimulus salience and mutual inhibition between unique elements.

When two compounds (AX and BX) are preexposed, the shared element X receives double the exposure of the unique A and B elements producing a larger reduction in X's salience due to stronger context associations and internal activation. Behaviorally, the effect can be manifest as latent inhibition, the phenomenon observed when exposure to a stimulus retards its ability to be associated with another later [19–21]. Due to the increased exposure to X, it suffers more latent inhibition than A (presumably through a loss of salience), allowing A to acquire greater associative strength than X during conditioning of AX, reducing generalization to BX [22,23]. However, in the Intermixed/blocked effect, X is exposed equally in both conditions and processes affecting only X cannot explain the effect.

Due to the within-compound associations proposed by [12] when AX and BX alternate, each presentation can evoke the representation of the other's unique element through their shared component X. The absent element, predicted but not present, comes to be inhibited by one present. Thus, in addition to A↔X↔B excitatory associations, A⊢B, inhibitory associations can form in the intermixed exposures. At test, inhibition between B and A can suppress the retrieval of A's representation and responses produced via A's prior conditioning, further reducing generalization.

An additional account can also be derived from the Mclaren and Mackintosh model [12]. Extended experience that produces strong within-compound associations may promote configural encoding. There, BX may become represented as a unique configuration rather than the sum of its elements, e.g., [13,24–26]. Such encoding would

make BX more distinct from AX and from its own components. That is, BX and AX may be represented simply as C and D, unique configurations that do not share common elements.

Hall [27] suggested a different mechanism, also based on inter-element associations, whereby the habituation or reduction in salience that occurs during pre-exposure is undone by the schedule in which the stimuli are pre-exposed. Hall discusses "perceptual effectiveness" as the ability of a stimulus to activate its representation. While habituation reduces perceptual effectiveness, Hall suggested that it can be restored when the stimulus is retrieved associatively in its absence. In Hall's theorizing, with alternating AX/BX exposure, A↔X↔B associations form so that X can retrieve A or B representations. On an AX trial the retrieved representation of B, in the absence of B itself, is said to restore the perceptual effectiveness of the physical B. That restored effectiveness increases its ability to affect responding evoked by X on a test; an effect with empirical support [3,4]. We will refer to this mechanism as restored perceptual effectiveness (RPE).

Some authors [1,28–31] suggested that good encoding of the unique elements is responsible for enhanced discrimination. For example, Lavis et al., [31] showed that additional exposures to unique elements A and B, along with intermixed AX/BX presentations, further enhanced the discriminability of the compounds. They argued that these additional exposures should further habituate the unique elements, producing a decrease in the AX/BX discrimination according to Hall´s account [27]. Rather, the additional exposures to A and B allowed for them to be better encoded, and produced attentional biases toward them [1,31].

Reduced generalization after exposure to only the test stimulus BX, prior to conditioning with AX, has received much less study [10,32–35]. As explanatory theories of discrimination are based on procedures involving exposure to both the conditioning and test stimuli, such an effect has implications. Sanjuán et al., [34] for instance, showed that rats pre-exposed to a compound flavor BX (B = 1% salt, X = HCl) displayed markedly reduced generalization of an aversion conditioned to a similar flavor AX (A = 10% sugar). This reduction was stronger than after exposure to B and X separately, showing that latent inhibition accrued to X alone cannot explain the effect. As only one compound was experienced, there is no opportunity for A⊣ B inhibition to form, removing that mechanism as an explanation as well.

With regards to a habituation/latent-inhibition-based account, Sanjuán et al., [34] observed a graded reduction in generalization as a function of the number of BX exposures (1 < 4 < 8). The course of conditioning of AX was also revealing. All the groups receiving pre-exposure to BX received the same overall exposure to X, varying only by whether that exposure was with B. One or 4 exposures to BX (7 or 4 exposures to X alone) produced a delay in conditioning with AX. That is, any latent inhibition accrued to X generalized to AX during its conditioning phase. Yet, when all exposures to X were with B, there was no generalization to AX. Latent inhibition, a mechanism that may contribute to changes in generalization was itself subject to generalization. Latent inhibition accrued to X during its exposure within BX did not generalize well outside of the compound.

The findings above suggest that BX became represented as a distinct unit, such as a configural cue. Yet, Sanjuán, et al., [34] showed that preexposure to BX reduced generalization from AX to BX more effectively than equivalent exposure to CX. Exposure to CX should produce a configural cue that is as dissimilar from AX as is BX. The effect's specificity to the test stimulus suggests that exposure modifies the internal representation of BX in a way that makes it perceptually distinct from related stimuli.

Similar reductions in generalization by test-stimulus exposure have been obtained in human participants (e.g., [35]), showing that experience with a single stimulus can improve subsequent discrimination between similar stimuli. Dwyer and Vladeanu [36], for example, found that simple exposure to a single face stimulus improved later discrimination of that face from similar ones. They interpreted this effect as reflecting more efficient encoding of the distinctive features of the exposed stimulus, indicating that direct experience with a single exemplar can sharpen its perceptual representation even in the absence of explicit comparison.

The enhanced encoding account outlined by [1,28–31] is applicable to this effect, as good encoding should occur whether or not one (e.g., BX) or more (AX/BX) appear. However, it may fare no better in accounting for the effect of BX

exposure. Exposure to BX may enhance the encoding of the now-unique element A during AX conditioning, reducing the amount that X conditions and reducing generalization to BX. Exposure to CX should have the same effect, and such an effect was not observed by Sanjuan, Alonso and Nelson [34]. The exposure to BX should also allow good encoding of B, however, exposure to BX was more effective than exposure to B and X separately, which should have been particularly effective in encoding B following the results of Lavis et al., [31] but exposure to BX was superior to that of the elements in reducing generalization.

The RPE mechanism proposed by Hall [27] can operate in an experiment designed to examine the effects of test-stimulus exposure as a similar stimulus must be conditioned with which to determine generalization. Exposure to BX alone offers little opportunity for restoration during preexposure, but recovery of perceptual effectiveness might occur later, during conditioning. When X is conditioned in AX+, it may retrieve B's representation, enhancing B's salience and, consequently, the discriminability of BX at test. Reduced responding to BX could be due to its perceptual difference, or the presence of a salient B disrupts any responding controlled by X (e.g., Pavlov's "external inhibition," [37]).

Another mechanism, yet to be explored in the context of perceptual learning, is retrospective revaluation (RR) which refers to a change in the associative status of a stimulus based on new learning about an associate [38,39]. For instance, in a human causal judgment task conducted by Chapman [38], participants initially experienced AB+/CD+ training where compounds of stimuli AB and CD were paired with an outcome (+) across several trials. In a subsequent phase, stimulus A alone was paired with the outcome (A+) to increase its associative strength, while stimulus C alone was presented without the outcome (C-) to decrease its associative strength. During the final test, stimuli B and D were presented. Stimuli B and D were not present during the second phase and no direct changes in their associative strength were expected. Nevertheless, B was rated as less effective than D in predicting the outcome. It is as if the associates of A (B) and C (D) were retrospectively re-valued in the second phase with their changes being the opposite of their associates.

There are theories for the RR phenomenon that the reader can pursue, e.g., [39,40], though they will not be discussed here as they are not being tested. Rather, we are assessing simply whether the phenomenon can operate in situations where the test stimulus is exposed and a similar stimulus subsequently conditioned. During BX preexposure, B and X can become associated. Later, when AX is conditioned, X can retrieve B's representation. According to retrospective revaluation explanations and empirical findings [38–49]. B's association with X's outcome would change in the opposite direction to that of X (i.e., a neutral B becomes inhibitory while X becomes excitatory) which would result in a reduced response to BX. This mechanism has not been explored in the context of perceptual learning.

The present experiments were designed to assess RR mechanisms contribution to enhanced discriminability following preexposure to a test stimulus, along with recovery of perceptual effectiveness as described by Hall [27]. If BX preexposure allows B to be retrieved during AX conditioning, B might (a) acquire inhibition toward the outcome, consistent with retrospective revaluation [39], or (b) recover its lost salience, increasing its perceptual effectiveness as proposed by Hall [27].

Interestingly, a single set of tests regarding inhibition can be used to separate these mechanisms. Rescorla [50] proposed that inhibition be diagnosed with two tests, a summation test, and a retardation test. In a summation test, a putative inhibitory stimulus (e.g., B) is paired with an excitatory stimulus with which it was not trained (Y). Responding to BY should be less than responding that to a compound which does not contain an inhibitory stimulus (e.g., DY). However, the summation test can be explained by the stimulus B commanding more attention, thereby disrupting responding. Thus, a retardation test was suggested to complement the summation test. There, B is conditioned and compared to the conditioning of a stimulus (D) that is not supposed to be inhibitory. If BX/DY differences on the summation test were produced by B being inhibitory of the outcome, conditioning of B should be slower than with D. Yet, if more attention is being devoted to B, then it should not only fail the summation test, B should condition faster than D.

If RR operates, then B should pass both summation and retardation tests. However, if RPE operates, then B should pass the summation test (i.e., external inhibition) and fail the retardation test. Experiment 1a and 1b used the same

participant pool and methods. Experiment 1a assessed the effects of pre-exposure to BX with a summation test, and Experiment 1b used a retardation test.

## 2. Experiments 1a and 1b

The experiments (Exp 1a and 1b) were conducted online using the Prolific Academic recruiting platform. Both used a video game, developed based on an earlier version [21], which has been shown to be an effective tool for investigating perceptual learning [35,51,52] and associative learning phenomena in general (e.g., [53–56]).

In the method, participants earn points by shooting torpedoes at a "cybergnostic space chicken" to pass a "promotion" exam. During the game, they can be attacked by the chicken with an "egg of destruction" which causes them to lose points. They are instructed that they can avoid point loss by suppressing their own rate of torpedo launching to save power prior to the attack. Colored sensors can be programmed to appear which signal the attack in the conditioning phases, and participants learn to suppress their mouse clicking (torpedo firing) during the presence of the sensor in anticipation of the attack to avoid point loss.

The design is shown in Table 1. The study employed a completely within-subjects design in which participants underwent two phases of training and a test. The first phase involved exposure to compound stimuli. An oval sensor was composed of two colors (e.g., BX), with the color represented by the first letter in a compound's name (e.g., B) occupying a small portion of the sensor, from left to right, while the color represented by the second letter occupying the remaining larger portion. Compounds BX, CY, and DZ were presented several times before the second phase which focused on conditioning with two additional stimulus compounds (AX and AY) each containing elements (X or Y) which had been associated with elements from pre-exposure. AX and AY were followed by the egg-of-destruction attack (+), representing the unconditioned stimulus (US). X should retrieve the representation of B, allowing the operation of RR or PE. Generalization and Summation tests were conducted in Experiment 1a. Retardation testing was conducted in Experiment 1b.

### 2.1. Experiment 1a

The effect of test-stimulus preexposure on generalization was assessed by comparing the BX compound with DY, a compound that also contains elements from previously exposed compounds, but has itself never been experienced. Based on previous findings with a similar method [35,51,52,56], we anticipated a perceptual-learning effect where the conditioning of AX would generalize less to BX than to DX, despite equal exposure to both B and D in compounds and equal experience with X.

The element B could be associatively activated during the AX trials, allowing retrospective revaluation to occur making B inhibitory. According to Hall [27] the effectiveness of stimulus B, which is otherwise being habituated, would also be

**Table 1. Design of Experiments 1a and 1b.**

| Pre-exposure 1a and 1b | Conditioning 1a and 1b | Testing 1a | Testing 1b |
|---|---|---|---|
| 6 BX/CY/DZ | 14 AX+/AY+ | Generalization Test 1 BX^/ DX^ | 8 BW+/DW+ |
| | | Summation Test 1 BY^/DY^ | |

*Note.* B, C, and D were smaller blue, yellow, and cyan portions on the left of a compound stimulus, counterbalanced. X, Y, and Z were larger green, brown, and orange portions on the right of the compound stimulus, counterbalanced. The small and large portions were counterbalanced against each other (36 orders in total). "A" was a small red portion on the left of the AX and AY compounds. "+" indicates an egg-of-destruction attack. "/" indicates intermixed trials. Numbers indicate the number of trials with each compound. "^" indicates a test-screen outcome. See text for details.

restored. The inhibitory properties of B were examined in a summation test where B was combined in compound with an excitor with which it was not trained (BY) and compared with a compound of DY. As D's associate, Z, was not experienced again after pre-exposure, there was no opportunity for either RR or RPE to operate. If B is inhibitory for the outcome (RR), or particularly salient (RPE), it should reduce suppression evoked by Y more than D. Suppression to BY (inhibitor + excitor) should be weaker than the suppression to DY (neutral + excitor). The same outcome on the BY/DY summation test would be expected based on Hall [27]. The restored effectiveness of B would make it functionally more salient and distracting on test relative to D, better reducing suppression controlled by Y by way of external inhibition [37].

The latent and mutual-inhibition mechanisms proposed by McLaren, Kaye and Mackintosh [11] and McLaren and Mackintosh [12] predict no effects here as the treatment of all the elements used in pre-exposure and testing was the same. If some configural process or unitization is responsible for the reduction in generalization to the test stimulus, we would again expect no differences on either test as BX and DX or BY and DY are equidistant from AX and AY, assuming they form unique cues. Moreover, all elements were equally exposed in compounds during pre-exposure, and never presented again until test. Thus, any account base on encoding or attention of these stimuli should be equivalent as the encoding experience was the same for all stimuli.

## 2.2. Experiment 1b

Experiment 1b was conducted using the same method and design in the pre-exposure and conditioning phases. Rather than summation and generalization tests, Exp 1b concluded with a retardation test where B and D were combined with W (white), on separate trials, and paired with an egg attack. During the summation test, B could inhibit suppression evoked by Y because it is inhibitory for the outcome (conditioned inhibition) by way of retrospective revaluation. In the case of B being inhibitory, acquisition of suppression in the retardation test should be *less* rapid in compounds involving B. Or, as mentioned earlier, B could affect X and Y by way of external inhibition. B could be especially salient and more distracting than D if B was associatively activated during AX conditioning and its effectiveness restored, as predicted by Hall [27]. In that case, a compound of BW should be more salient than DW and acquire suppression *more* rapidly. As outlined for Experiment 1a, no differences are expected based on the other theories discussed in the introduction due to the equivalent treatment of the physical stimuli in the pre-exposure phase and their absence in conditioning.

## 3. Method

### 3.1. Participants

All procedures involving human participants and their data were reviewed and approved by the ethics committee for research involving human subjects of the University of the Basque Country (CEISH-UPV/EHU), BOPV 32, 17/2/2014. Experiment 1a was run on Feb 26th, 2024. Experiment 1b was run on May 20th, 2024. Experiment 1a was covered by a multi-year approval issued on Oct 17, 2019 (M10_2019_2011), for the project *The prediction error and processing of context*. Experiment 1b was covered by a renewal and extension of that protocol on May 2, 2024 (M10_2024_079), for the project *Extinction, inhibition and representational changes in contextual control.* Both studies were conducted in accordance with the ethical standards of the Helsinki Declaration and its later amendments.

Participants were volunteers from Prolific Academic. As anonymous members of research-participant platform [57] they were aware that they are participating in research when they selected the task for which Prolific has informed them that they qualify. The study description informed potential volunteers of the requirements of the research project (a computer capable of conducting the task), that their participation was voluntary, and that they could withdraw their data from the project after completion. They were explicitly informed that by clicking the link which directs them to the experiment they were providing their consent to participate. There are no minors participating in the Prolific platform.

For sample-size determination we considered that we used the exact apparatus as used by Nevado and Nelson [55] to investigate the "renewal effect" (e.g., [54]). Extinguished conditioned responding increases with a change in the

background context where extinction occurred. In that online within-s experiment Nevado and Nelson obtained $\eta^2_p = .25$ for the renewal effect, which is our current best estimate as to the maximum effect we might expect to observe. Counterbalancing the different identities of the stimuli required 36 different combinations. The method updates a log when each participant is assigned, keeping conditions balanced. However, participants may get assigned but not complete the task, resulting in the need to recruit again. We planned to recruit 100 participants to ensure that we would fill all 36 conditions at least twice through random assignment, and have adequate power. Power was .17, .70, and .98 to detect Cohen's [58] small, medium, and large effects, and .8 to detect $\eta^2_p = .075$, (one-third of the estimated maximum effect). The effect observed in Experiment 1a produced $\eta^2_p = .05$. For Experiment 1b N was increased to plan on recruiting 150 participants to have a power of .8 to detect that effect size.

Participants were randomly assigned to counterbalancing conditions without replacement until all conditions were filled and then re-entered into the pool. Participants might begin the experiment and be assigned, updating the pool, but not complete the study. For that reason, more slots than needed were available resulting in the recruitment of 102 people in Exp 1a and 165 in Exp 1b. The age and gender statistics of the sample, based on those who reported such demographics through Prolific, are shown in Table 2. The distribution of gender was independent of the experiment, $X^2 = .34$, $p = .56$.

### 3.2. Apparatus

The video game used in the experiment has been used before, and detailed descriptions can be found there [55].

The apparatus is shown in Fig 1. Participants played the video game from a first-person view as if they were flying in a spaceship within a colorful 3D galaxy. The interface featured a dark gray gun attached to a pillar at the top of the screen. The gun tracked a red circular crosshair which was moved by the participant's mouse. Key information, such as the current context's name (Boutonia, which was not manipulated in the current experiments), was displayed in a translucent label on the top left of the screen. Another panel located at the top right of the screen displayed either "Gaining Points," "Not gaining points," or "_____ points lost" (with the number of points lost displayed in the blank). Text instructions (detailed in [55]) were presented on a translucent black panel that rose from the bottom of the screen and through a pre-recorded male voice.

Participants began by entering their Prolific ID and pressing "B" to start the game, which commenced in a "training context" that appeared as a green wireframe cube. Participants were guided through a brief tutorial that explained how to operate the gun, charge the weapon by clicking the mouse at a rate of three clicks per second, and maintain a steady rate of firing at the chicken.

The tutorial emphasized the importance of interpreting in-game events, such as flashing lights or sounds, to predict "egg of destruction" attacks by the chicken. Participants were instructed to reduce their firing rate when anticipating an attack to conserve power and minimize point loss. Following the instructions, participants completed a short multiple-choice quiz designed to ensure their understanding. Questions were repeated until answered correctly, with

**Table 2. Sample demographics.**

| Experiment | Reported Gender | N Reporting | Min. age | Max age | Mean | Sd |
|---|---|---|---|---|---|---|
| 1A | Male | 62 | 20 | 80 | 41.29 | 14.13 |
| | Female | 40 | 19 | 68 | 37.05 | 13.41 |
| 1B | Male | 103 | 18 | 73 | 37.61 | 12.48 |
| | Female | 57 | 19 | 66 | 39.92 | 12.60 |

*Note.* Table shows the distribution of self-reported gender by experiment. It also reports the mean, standard deviation, minimum and maximum values for age.

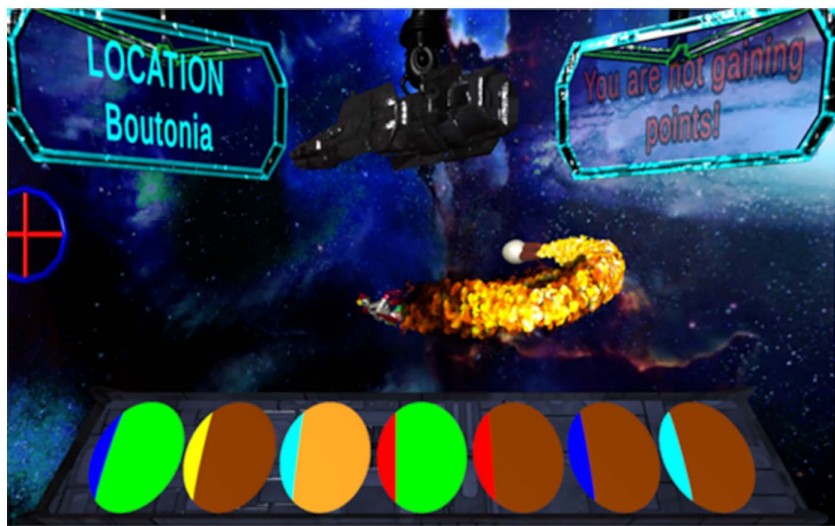

**Fig 1. Screenshot of the apparatus.** Figure 1 shows the apparatus and example stimuli, from left to right, BX, CY, DZ, AX, AY, BY, DY. Blue, Yellow and Cyan were counterbalanced as B, C, and D. Orange, Green, and Brown were counterbalanced as X, Y, and D. A was red. Though shown in different positions here, all CSs were presented in the middle position (fourth from the left). In the image, the Chicken has launched an egg of destruction (the US).

incorrect responses prompting a "wrong answer" message in red and a repeat of the question. Correct answers were followed by a feedback screen indicating the answer was correct and reiterating why the answer was correct.

After the tutorial a robotic chicken appeared together with a music background. The chicken was animated with visual and audio features, including randomly varying movement (speed and direction) and randomly varying robotic chicken sounds. The randomness of the chicken's behavior ensured unpredictable relations between the chicken's behavior and other aspects of the design. The view of the background changed as the camera tracked the movement of the chicken in 3d space.

During the game, eight different "Sensor" stimuli were used, consisting of color combinations flashing (fading on and off, cycling at 3 Hz) on the center oval of the seven black ovals appearing in the trapezoidal panel the bottom of the screen. The unconditioned stimulus (US) was an "Egg of Destruction." This egg was visibly launched from the rear of a chicken, accompanied by a burst of flame. The egg, propelled by flames from two attached exhausts, traveled rapidly until it struck the participant's screen, resulting in a large fireball explosion accompanied by a deep explosion sound and electrical after effects. The time from the egg's launch to its impact varied slightly depending on the chicken's distance from the screen, but the entire sequence, from launch to explosion, was completed in no more than 4 seconds. The attack drained up to 100 points based on "damage" which was a positively accelerating function of their suppression during the preceding sensor (see [55], for a full description of the mapping of suppression to damage). No suppression would result in maximum damage (d = 1) and a loss of 100 points (i.e., d x 100), while complete suppression would result in no damage (d = 0) and no loss of points.

A "test screen" could be presented which was a 5-s screen of static with the words ""Test control has no information on the outcome at this time. Simulation will resume shortly." overlaid. The text appeared in the center of the screen in a blue font. This screen was used as the outcome on test trials to help avoid carry-over effects in testing [59,60]. The screen also appeared in training, as indicated in the procedure below, to familiarize the participants with its occurrence.

Sensor colors were composed of a left-most (left 16% of the sensor area) and right-most (right 84% of the sensor area) color. The colors appearing in the left-most of the sensor were B, C, D, and A while X, Y, and Z colors were in the

 

right-most part. "A" was always Red, "W" was always white. Elements B, C, and D were Blue, Yellow, and Cyan, counterbalanced (6 combinations). Colors X, Y, and Z were Green, Brown, and Orange, counterbalanced (6 combinations), and counterbalanced fully against B, C, and D (36 total combinations). When presented, the sensors always appeared in the fourth from the left of the seven ovals. Examples are shown in Fig 1, though all stimuli appeared in the center oval in the experiments.

Stimuli A and W were not counterbalanced with the other colors as neither was ever compared to any other. Only critical stimuli being compared were counterbalanced.

### 3.3. Procedure

**3.3.1. Experiments 1a and 1b.** Following the instruction phase, the chicken and player were pulled through a "wormhole" into a colored galaxy where the experimental manipulations took place. Participants engaged in three experimental phases: Exposure, Conditioning, and Testing. During the Exposure phase, BX, CY, and DZ were presented one time each in each of 6 blocks. Each presentation lasted 5 seconds and the order of presentation within a block was randomized. There was no outcome following the presentations, except on block 3 the test-screen outcome appeared following each stimulus.

Participants were firing at the chicken during this phase and all subsequent phases. No other specific events occurred in phase 1. The inter-trial interval (ITI) was randomly determined on each trial in each phase from a flat distribution with a mean of 11 seconds and a standard deviation of S = 5 and clipped to a range of 5–17 (resulting $= 10.98$, S = 4.57).

The conditioning phase began uninterrupted. Colored sensors AX and AY each appeared 14 times and were each paired with an "Egg of Destruction" attack. There were 7 blocks of four trials containing two AX and two AY trials in each block, with the trial order randomly determined. One trial with both AX and AY was followed by the test-screen outcome on both the 2nd and 6th blocks to ensure that the test screen was not associated with any particular phase.

**3.3.2. Experiment 1a: Compound Testing.** Compound tests were used to assess generalization and inhibition. Generalization from AX was assessed with a single presentation of BX and one of DX, each followed by the test-screen outcome. The order of the two trials was randomized. The inhibition assessment was a summation test consisting of a single presentation of BY and one of DY. The order of the two trials was randomized. The order of the summation and generalization tests was randomized.

**3.3.3. Experiment 1b Retardation testing.** Experiment 1b followed conditioning with a retardation test where B and D were combined with W. We assumed that simply using B and D with the remaining portion of the sensor black may promote attention to these stimuli obscuring any differences that the pre-exposure manipulation may have produced. There were 16 trials arranged in four blocks of four trials. Each block contained 2 BW+ and 2 DW+ trials with the order of the four trials randomly assigned.

### 3.4. Data analysis

The number of mouse clicks during the ITI and CS were recorded. The method was modeled after the conditioned-emotional-response procedure where lever-pressing rats receive CS-Shock parings and freeze during the CS. Those studies have a long history (e.g., [61]) of using a suppression ratio as the measure of response suppression, which helps correct suppression for individual differences in baseline response rates. The suppression ratio is the measure that has been used in this method and other similar ones [51–55] and was used here. The ITI and CS responses were converted to rates (clicks per second) and transformed to standard suppression ratios as CS rate/ (ITI rate + CS rate). Suppression was analyzed using repeated measures Analysis of Variance (ANOVA) and Type III sums of squares by IBM SPSS v23 [62]. Effect sizes are reported as partial eta squared ($\eta^2_p$) and 95% confidence intervals were calculated using non-central F distribution functions for Excel found in Nelson [63].

Participants suppress very little during the first second of the CS presentation in this method online, and suppression tends to asymptote around .2 (see [55,59] c.f., [54]). To increase the sensitivity of the measure, we analyzed suppression on a per-second basis of each second of the CS. In the method upon which the present method was conceptually modeled, suppression during the CS has been shown to increase over the duration of the CS revealing effects that otherwise may go undetected [64].

The first two phases were analyzed with 1a and 1b combined, and "Experiment" (1a vs 1b) included in the analysis. The test data were analyzed by experiment. Suppression can only be calculated when ITI rates of responding are above zero. The occasional zero ITI rate was replaced with the mean of the participant's other non-zero trials of the phase. We followed previous procedures used with this method [55] and removed participants from the analysis of a phase when 20% or more trials were replaced in that phase. Error bars in the figures show the standard error of each data point.

### 3.5. Transparency

The data are available at the Center for Open Science repository under the projects managed by James Byron Nelson (**10.17605/OSF.IO/TYMUB**) in .xlsx format for Excel. The method will be made available upon request to either author explaining its intended use.

## 4. Results

There were no differences between Experiments 1a and 1b in the training data, which are combined and shown in Fig 1. The figure shows suppression on each second of the presentation of each CS. Data from pre-exposure are shown at left above "Exposure" and conditioning of AX and AY are shown above "Conditioning." (Fig 2).

### 4.1 Exposure

Data from exposure were analyzed with a CS (BX, CY, DZ) x Trials x Seconds x Experiment ANOVA. No participant was removed by the ITI-rate screening. The analysis showed a Trials x Seconds interaction, $F(20, 5280) = 1.6$, $p = 0.04$, $MSE = 0.03$, $\eta^2_p = 0.006$, $CI_{95\%} = 1.9 \times 10^{-8}$ - 0.007. Inspection of the figure suggests that suppression decreased during the

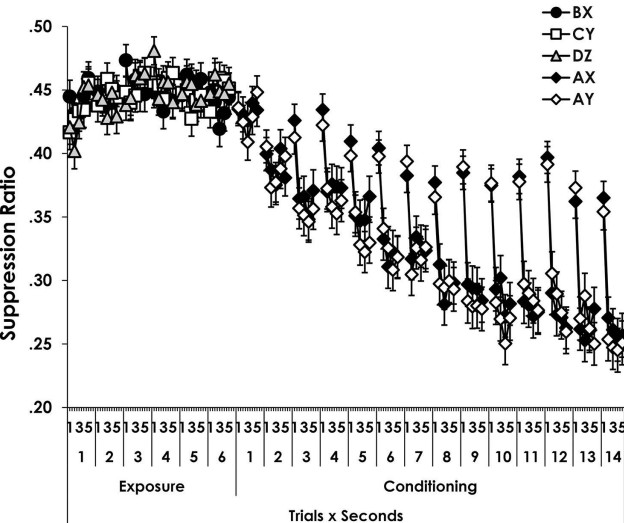

**Fig 2. Suppression during pre-exposure and conditioning.** Figure 2 shows suppression to the stimuli on each second of each trial during exposure and conditioning for Experiments 1a and 1b combined. Notice that the Y axis begins at .2. Error bars show the standard error of the mean.

seconds of the first trial, but randomly varied among other trials. Of most importance, there were no effects involving CS or Experiment. The Trials x Seconds effect reported above was the only reliable effect in the analysis, $ps \geq .11$.

## 4.2. Conditioning

Data from conditioning were analyzed with a CS (AX vs. AY) x Trials x Seconds x Experiment ANOVA. Six participants were removed by the ITI screening. The analysis showed effects of Trials and Seconds which were superseded by a Trials x Seconds interaction, $F(52, 13416) = 4.81$, $p = 2.4 \times 10^{-27}$, $MSE = 0.03$, $\eta^2_p = 0.02$, $CI_{95\%} = 0.011–0.02$. The pattern is apparent in the figure. Suppression was acquired over trials, and increased across seconds within each trial. The increase along seconds within a trial was more evident in later trials than in earlier ones. There were no other effects, $ps \geq .17$.

## 4.3. Compound Testing (1a)

The data from the Compound Testing are shown in Fig 3. Compounds assessing generalization (BX and DX) are shown at left and those assessing inhibition (BY and DY) are shown at right. Suppression on each second of the CS presentation is presented.

The data were analyzed with a Compound-type (Generalization involving X vs. Summation involving Y) x Revaluation (B vs D) x Seconds ANOVA. Five individuals were eliminated by the ITI screening. Either an RR or RPE effect would appear as a reduction in suppression to BX relative to DX and a reduction in suppression to BY relative to DY. A lack of a retrospective-revaluation effect would produce either no effects, or no effect with BY and DY, thus producing an interaction of Compound-Type with Revaluation. The analysis produced only a Revaluation main effect, $F(1, 96) = 5.07$, $p = 0.03$, $MSE = 0.08$, $\eta^2_p = 0.05$, $CI_{95\%} = 1.0 \times 10^{-6} - 0.16$. There was no effect of Compound-Type, $F < 1$, and no Compound-Type x Revaluation interaction, $F(1,96) = 1.24$, $p = .27$. There was an effect of seconds, $F(4, 384) = 11.97$, $p = 3.5 \times 10^{-9}$,

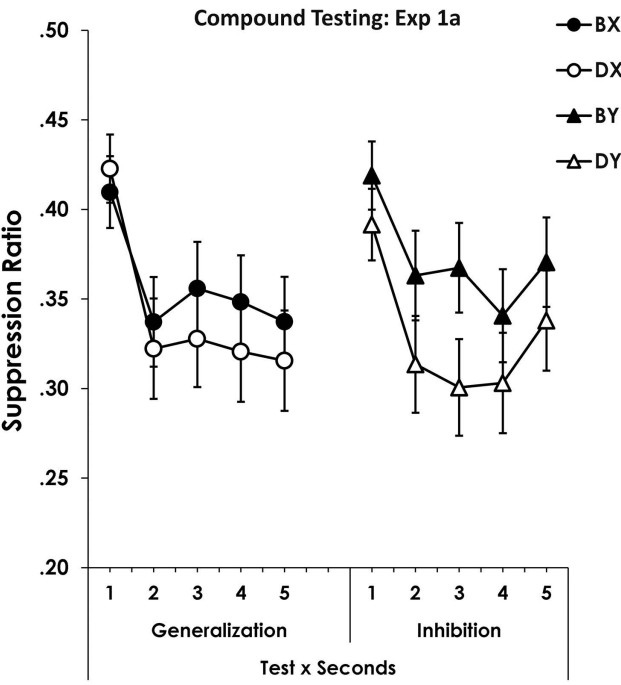

**Fig 3. Generalization and summation testing.** Figure 3 shows suppression to BX and DX in the generalization test at left and BY and DY from the inhibition testing at right. Error bars are omitted to remove clutter. Notice that the Y axis begins at .2. Error bars show the standard error of the mean.

$MSE = 0.04$, $\eta^2_p = 0.11$, $CI_{95\%} = 0.05$–$0.17$, but no interactions of seconds with the other variables, $ps \geq .41$. Testing with the exposed CS (B), whose associate had been conditioned, led to a greater disruption of responding in a compound than did D that had been exposed and whose associate was not experienced again.

### 4.4. Retardation Testing 1b

The retardation test data are presented in Fig 4 which shows suppression on each second of each trial to the BW and DW stimulus compounds. The right-most portion shows the effects of Trials and Compound type, collapsed over seconds.

The data from retardation testing were analyzed with a Compound Type (BW vs. DW) x Trials x Seconds ANOVA. Ten persons were excluded by the ITI response screening. The analysis showed a main effect of Compound Type where suppression to BW was slightly greater than DW, $F(1, 150) = 8.12$, $p = 0.005$, $MSE = 0.06$, $\eta^2_p = 0.05$, $CI_{95\%} = 0.005$–$0.13$. There were effects of Trials and Seconds which were superseded by a Trials x Seconds interaction, $F(28, 4200) = 2.8$, $p = 1.3 \times 10^{-6}$, $MSE = 0.02$, $\eta^2_p = 0.02$, $CI_{95\%} = 0.006$–$0.02$. The Trials x Seconds effect is seen in the figure where suppression increased across seconds of the CS presentation, with that effect being more pronounced later in training. There were no further effects, $ps \geq .64$.

## 5. General discussion

The study investigated whether retrospective revaluation (e.g., [39]) or a recovery of perceptual effectiveness (e.g., [27,65]) can contribute to the enhanced discriminability observed when a test stimulus has been pre-exposed. In both

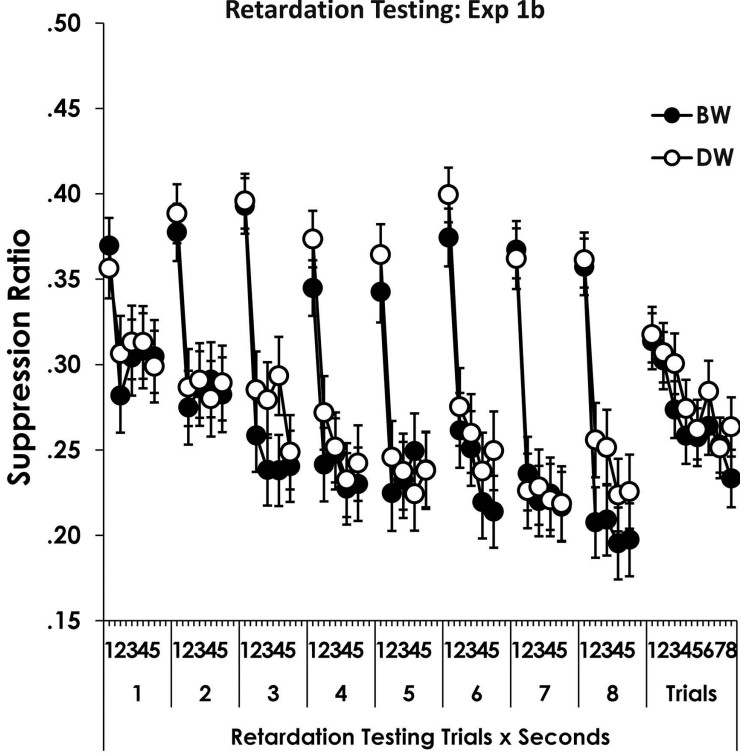

**Fig 4. Retardation testing experiment 1b.** Figure 4 shows suppression (Y axis begins at .15) to BW and DW on each second of each trial above *Retardation Testing Trials x Seconds*, and shows suppression averaged over seconds on each trial above *Trials*. Error bars show the standard error of the mean.

experiments experimental (B) and control (D) elements were pre-exposed in compounds (BX and DZ). Following pre-exposure, compounds AX and AY were conditioned. The compounds were equivalent in that they each contained a new element (A) and one that had been presented in a prior-exposed compound (X from BX, and Y from CY). Conditioning with AX should allow for the associative activation of B which may make it inhibitory by way of RR, or have its otherwise habituated effectiveness restored (RPE). Neither of these processes can operate on D for which its associate Z is never encountered again.

In Experiment 1a, B's associate X was tested in compound with B, matching the pre-exposed BX, or it was tested with D, an equally pre-exposed stimulus, but one which produced a novel compound. B and D were also tested with Y, which had been pre-exposed with C and conditioned with A, matching its history with X, but neither test compound matched pre-exposure. If B were inhibitory through RR, or more perceptually salient through RPE, there should be less suppression to BX than DX, as well as less suppression to BY than DY. The results were simple to interpret; the effects were dependent on only B. There was an effect of whether an element of the test stimulus could have been associatively activated earlier, producing either RR or RPE, that applied equally to tests involving the pre-exposed test stimulus itself, or tests involving new BY compounds as indicated by the lack of an interaction [66,67].

The retardation test of Experiment 1b determined whether the mechanism of action observed in 1a was RR or RPE. An RR interpretation would assume B was inhibitory for the outcome, thus a compound of BW should condition more poorly than DW. A RPE interpretation would assume the more effective representation of B might commanded more attention than D, thus a compound of BW would condition better that DW. Though the effect was small, the results were again direct. Conditioning with BW was more effective than with DW, supporting the idea that B was effectively more salient than D, producing a more salient BW compound than DW. The results support RPE as they are consistent with the idea that B was more salient than D, allowing it to pass a summation (Experiment 1a) test, while failing a retardation (Experiment 1b) tests for inhibition.

Interestingly, a recovery of perceptual effectiveness through associative activation differs from retrospective revaluation in that it places the effect in perceptual terms, rather than in terms of the associative status of a stimulus. The activation of B by X during AX conditioning might restore a representation of B that had diminished in salience during pre-exposure. The perceptual system, in this case, could treat the associative activation of B as a signal to "refresh" or recover the stimulus' prominence, making it more effective in subsequent presentations, such as in the summation test. This restoration of perceptual salience may be an attentional process whereby the absence of otherwise represented stimuli redirects attention to those stimuli, rather than a process of associative learning that modifies the stimulus's connection to an outcome (either through excitation or inhibition). In this way, the explanation is similar to the enhanced encoding suggested by [1,28–31] only here the enhancement comes from experiencing the representation indirectly, rather than through the presence of the stimulus itself. Salience recovery may be a consequence of associations, but serves as an adjustment that enhances the influence of the stimulus in future encounters, while leaving its associative connections unchanged.

Differentiation between similar stimuli would not require that one of the stimuli undergoes conditioning to acquire meaning, as required by an RR interpretation. This point is particularly relevant as human studies, to the extent that simply noticing differences is not reinforcing [68], conditioning is not present. Instead, simply encountering a stimulus may be sufficient to associatively activate the distinctive elements of similar stimuli and restore or enhance their perceptual salience. These perceptually effective unique elements enable the organism to discriminate between the stimuli. In this respect, our interpretation aligns completely with Hall's [27] proposal and evidence showing that activation of the unique element's representation by the shared element can affect its salience. The proposal expands Hall's analysis in that Hall's discussion of this process focuses exclusively on preexposure and research exploring this idea has evaluated conditions where activation of the unique element of a stimulus is produced during preexposure. Our research here shows that this salience modification can occur not only during preexposure, but at any moment when there is an opportunity to reactivate the unique features of the stimuli.

The presence of B, when conditions would permit its associative activation during conditioning of AX, reduced generalization from AX to BX relative to that produced by an equally exposed cue, D, whose associate was not conditioned. B also reduced generalization from AY to BY better than D, and a compound of BW conditioned better than did a compound of DW. These findings suggest that B was more effectively salient than D, producing more distraction or "external inhibition" on the BX test, and was easier to learn about during the retardation test.

These findings contribute to the broader literature on perceptual learning and associative learning. While previous studies have emphasized the role of alternating exposure to stimuli in promoting perceptual learning, the present results further demonstrate that exposure to a single stimulus compound (BX) is sufficient to reduce generalization in well-controlled within-subject designs. Perceptual learning is multiply determined. Mechanisms identified in the introduction that deal with differential latent inhibition to common and unique elements [22,23], mutual inhibition between unique elements [11,12], or differential attention to unique elements [1,28–31] that can form with exposure to both training and test stimuli undoubtedly contribute to perceptual learning effects. In the present research, those were either held constant (latent inhibition, degree of encoding/attention) or were unable to form (mutual inhibition), thus the effect observed here cannot be explained by those mechanisms. The absence of those mechanisms may account for the small effect that our well-powered within-subject designs demonstrated, further reinforcing that a change in salience produced by associative activation is a contributing factor to perceptual learning, but certainly not the only factor.

How do we recognize that a new coffee is different from one previously experienced? A coffee has many flavors within it. Some of those flavors, such as a degree of bitterness, are common to all coffees while other flavors, such as nutty or chocolate notes, are unique to different coffees. Experience with a coffee establishes associations between those elements. When a new coffee is experienced, the common element retrieves a representation of the unique element from the first coffee, bringing the distinctive features of the first coffee back into focus. It would be at this moment that our perceptual system achieves differentiation between the two coffees. This process might enhance the perceptual effectiveness of those unique features, making discrimination even simpler with later encounters. The distinctive characteristics of the first coffee might become more salient, allowing them to be better compared with other coffee attributes.

## Author contributions

**Conceptualization:** Maria Del Carmen Sanjuan, James Byron Nelson.

**Data curation:** Maria Del Carmen Sanjuan, James Byron Nelson.

**Formal analysis:** Maria Del Carmen Sanjuan, James Byron Nelson.

**Funding acquisition:** Maria Del Carmen Sanjuan, James Byron Nelson.

**Investigation:** Maria Del Carmen Sanjuan, James Byron Nelson.

**Methodology:** Maria Del Carmen Sanjuan, James Byron Nelson.

**Project administration:** Maria Del Carmen Sanjuan, James Byron Nelson.

**Resources:** Maria Del Carmen Sanjuan, James Byron Nelson.

**Software:** James Byron Nelson.

**Supervision:** Maria Del Carmen Sanjuan, James Byron Nelson.

**Validation:** Maria Del Carmen Sanjuan, James Byron Nelson.

**Visualization:** Maria Del Carmen Sanjuan, James Byron Nelson.

**Writing – original draft:** Maria Del Carmen Sanjuan, James Byron Nelson.

**Writing – review & editing:** Maria Del Carmen Sanjuan, James Byron Nelson.

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
