## [Decision Letter · Decision Letter 0]

8 Sep 2025

PONE-D-25-32174

Perceptual learning mechanisms with single-stimulus exposure

PLOS ONE

Dear Dr. SANJUAN ARTEGAIN,

Thank you for submitting your manuscript to PLOS ONE. I am very grateful to the reviewer who gave such thorough feedback (below). Unfortunately one other reviewer was not able to complete their review, but I have myself had a careful read of the manuscript.  After careful consideration, we feel that it has merit but does not fully meet PLOS ONE’s publication criteria as it currently stands. Therefore, we invite you to submit a revised version of the manuscript that addresses the points raised during the review process.

In general the introduction should be more succinct, currently it is far too long and unwieldy. Please follow the reviewers advice on what can be left out and what should be included, I agree with these comments.

As indicated by the reviewer, error bars should be included and within-subject error bars are particularly useful in this context. You can easily calculate within-subject error bars, see this link for a very useful step by step guide: https://www.cogsci.nl/blog/tutorials/156-an-easy-way-to-create-graphs-with-within-subject-error-bars

It is a concern that colours were not fully counterbalanced - please explain this decision in the methods and  acknowledge this as appropriate throughout the manuscript.

We look forward to receiving your revised manuscript.

Kind regards,

Poppy Watson

Academic Editor

PLOS ONE

Journal Requirements:

[No competing interest to declare].

4. Please amend the manuscript submission data (via Edit Submission) to include author María del Carmen Sanjuán.

5. Please amend your authorship list in your manuscript file to include author MARIA DEL CARMEN SANJUAN ARTEGAIN.

6. Your abstract cannot contain citations. Please only include citations in the body text of the manuscript, and ensure that they remain in ascending numerical order on first mention.

7. Your ethics statement should only appear in the Methods section of your manuscript. If your ethics statement is written in any section besides the Methods, please delete it from any other section.

8. We note that Figure 1 in your submission contains copyrighted image. All PLOS content is published under the Creative Commons Attribution License (CC BY 4.0), which means that the manuscript, images, and Supporting Information files will be freely available online, and any third party is permitted to access, download, copy, distribute, and use these materials in any way, even commercially, with proper attribution. For more information, see our copyright guidelines: http://journals.plos.org/plosone/s/licenses-and-copyright.

Reviewers' comments:

Reviewer's Responses to Questions

**Comments to the Author**

1. Is the manuscript technically sound, and do the data support the conclusions?

Reviewer #1: Partly

2. Has the statistical analysis been performed appropriately and rigorously? 

Reviewer #1: Yes

3. Have the authors made all data underlying the findings in their manuscript fully available?

Reviewer #1: Yes

4. Is the manuscript presented in an intelligible fashion and written in standard English?

Reviewer #1: Yes

5. Review Comments to the Author

Reviewer #1: This study tests two explanations for the phenomenon that preexposure to a compound test stimulus (BX) alone suffices for learning (or reduces generalized aversion), after conditioning with another stimulus, AX. The mechanisms considered are retrospective reevaluation (RR) and restoration of perceptual effectiveness (RPE). Under RR, B comes to inhibits the outcome; under RPE, B becomes more salient or distinct from the conditioned compound. Two experiments were conducted using dichromatic symbol compounds in a video game, with suppression of a shooting response as the dependent measure. In both experiments, compounds BX and DX were pre-exposed, followed by conditioning with AX and AY. In Experiment 1, a summation test was conducted with compounds BY and DY, and BY passed the test. In Experiment 2, a retardation test was conducted with compounds BW and DW, and BW failed the test. The authors conclude that the results support RPE - or enhanced salience of B - as the learning mechanism.

This could be an interesting and useful paper provided some changes are made.

Major comments

1. Length and clarity of introduction: The introduction should be reduced substantially. At present, it reads more like the first draft of a master’s thesis than a manuscript for publication. The phenomenon being studied and the competing hypotheses need to be explained earlier and more clearly and concisely. The purpose of the study should be stated in 4-5 pages total, focusing directly on: the effect of interest, the competing explanations (RR vs. RPE), whether either experiment is a replication of earlier work or whether both are novel. Other details - Gibson’s ideas and mediated learning, for example - can be omitted.

2. Relation to human perceptual learning literature: The study needs to be more clearly situated within the context of human perceptual learning, in addition to or instead of the animal learning studies. For example, Lavis et al (2011) showed that exposure to a single element can enhance discrimination. Is this similar to the effect reported here? If not, please clarify the differences. Mitchell & Hall (2014) provide a review of the issues raised here, and other work by Dwyer’s group also should be cited.

3. Configural hypothesis (sections 2.1 and 2.2): The description of the configural hypothesis is not clear enough. Please state explicitly what predictions this hypothesis makes for each experiment, and how those predictions differ from the RR and RPE accounts. Please add all predictions to Table 1.

4. Stimulus salience: Cue A was always red, which is a particularly salient colour. This raises the question of whether the effects reported might have been different if A were less salient. The colour of A should have been counterbalanced across subjects. Please discuss this as a potential limitation of the results.

Further comments:

Abstract and general

The purpose of CY is unclear. If CY wasn’t relevant to these findings, please remove it from the abstract and elsewhere.

Introduction:

- Pages 10-11, lines 205-218: confusing and misplaced. If the results can be explained ‘without appeal to configuration’, why not say this earlier where you explain latent inhibition? Please clarify.

- Page 11, line 223 onwards: confusing. Does blocking contradict the configural explanation? If not, why isn’t the configural account sufficient?

- Page 13, line 270 onwards: When AX and BX are exposed in alternation, you state that A and B’s effectiveness is recovered. This seems the opposite of the mutual inhibition proposed by McLaren. Please explain.

- Page 15, lines 305 onwards: In the food+tone example, please add the compound stimulus notations (BX, AX etc) so as to compare with previous examples

- Page 15 lines 314 onwards: from here onwards, I ceased to follow the logic. Please delete, or reduce and simplify.

Methods

- The description of methods and the experiment (pages 18-28) is repetitive and could be shortened

- Table 1 should include predictions for both experiments under RR, RPE, and the configural hypothesis

- Page 20, line 426-428: The sentence “…better reducing suppression controlled by X by way of inhibition” is difficult to understand. Please rephrase.

Results

- Please explain the suppression ratio measure in a bit more detail

- Figures 2-4. Error bars should be shown regardless of clutter. To reduce clutter, you could add them to the first and last symbols only. Or, you could plot within-subjects error bars (Cousineau, 2005), which would be appropriate here and typically smaller.

- Experiment 1a: Lines 644-655 say that passing the summation test indicates RE, but the discussion says it could indicate RE or RPE. Please clarify.

- Experiment 1a: Please comment on performance differences between BX vs. BY. It appears suppression was different in those conditions. Does this have any implications?

- Experiment 1b: The main effect of compound type is small, and likely emerges because of the power of the repeated-measures analysis. The discussion describes this as a clear effect, which is overstated. Please modify.

Discussion

- Expand the discussion of human perceptual learning studies (e.g., refs below)

- Discuss the potential caveat of red A's salience

- Page 34, line 709 onwards: The explanation of salience modulation is unclear. Please clarify

- Page 35: This section is unclear and redundant with previous sections and not well tied to human learning studies

- Page 36, lines 746-748: What is the evidence that “…differential latent inhibition…undoubtedly contribute[s] to perceptual learning effects” in humans?

References

1. Lavis, Y., Kadib, R., Mitchell, C., & Hall, G. (2011). Memory for, and salience of, the unique features of similar stimuli in perceptual learning. Journal of Experimental Psychology: Animal Behavior Processes, 37(2), 211.

2. Mitchell C & Hall G. (2014). Can theories of animal discrimination explain perceptual learning in humans? Psychological Bulletin, 140(1), 283-307.

3. Cousineau D. Confidence intervals in within-subjects designs: A simpler solution to Loftus and Masson’s method. Tutorial in Quantitative Methods for Psychology 2005;1(1):4–45.

6. PLOS authors have the option to publish the peer review history of their article (what does this mean? ). If published, this will include your full peer review and any attached files.

**Do you want your identity to be public for this peer review?** For information about this choice, including consent withdrawal, please see our Privacy Policy .

Reviewer #1: No

---

## [Author Response · Author response to Decision Letter 1]

14 Nov 2025

Please see Responses_to_Review.docx.

---

## [Decision Letter · Decision Letter 1]

18 Dec 2025

PONE-D-25-32174R1Perceptual learning mechanisms with single-stimulus exposurePLOS One

Dear Dr. SANJUAN ARTEGAIN,

Thank you for submitting your revised manuscript to PLOS ONE and for clarifying some of the earlier points. I am pleased to say that your manuscript will be accepted for publication after some minor edits are made. Please see the comments of the reviewer below - it is also very important that you carefully read the manuscript for small grammatical errors because PLOS ONE does not copyedit manuscripts before publication.

We look forward to receiving your revised manuscript.

Kind regards,

Poppy Watson

Academic Editor

PLOS One

Journal Requirements:

Reviewers' comments:

Reviewer's Responses to Questions

**Comments to the Author**

1. If the authors have adequately addressed your comments raised in a previous round of review and you feel that this manuscript is now acceptable for publication, you may indicate that here to bypass the “Comments to the Author” section, enter your conflict of interest statement in the “Confidential to Editor” section, and submit your "Accept" recommendation.

Reviewer #1: (No Response)

2. Is the manuscript technically sound, and do the data support the conclusions?

Reviewer #1: Yes

3. Has the statistical analysis been performed appropriately and rigorously? 

Reviewer #1: Yes

4. Have the authors made all data underlying the findings in their manuscript fully available?

Reviewer #1: Yes

5. Is the manuscript presented in an intelligible fashion and written in standard English?

Reviewer #1: Yes

6. Review Comments to the Author

Reviewer #1: The revision of the manuscript is adequate and the authors have addressed most comments. A few remaining points:

1. Please correct grammatical errors and typos throughout. (e.g., line 87 promotes; line 100 awkward; line 190 that; line 542 might commanded, line 567 awkward clause at end of sentence; several more in discussion and other parts of the manuscript)

2. Line 151: The RPE is mentioned here for the first time, without explanation or prior definition (abstract doesn’t count). Please fix.

3. Figure captions for figures 3 and 4 say error bars are omitted, even though error bars are now included.

4. Around line 534, the authors could acknowledge that the effect of compound type in Experiment 1b is small (in addition to ‘direct’) to cohere better with the results section (which says ‘slightly greater’), and lines 596-97 in the discussion, where potential reasons are given for the small effect size.

7. PLOS authors have the option to publish the peer review history of their article (what does this mean? ). If published, this will include your full peer review and any attached files.

**Do you want your identity to be public for this peer review?** For information about this choice, including consent withdrawal, please see our Privacy Policy .

Reviewer #1: No

---

## [Author Response · Author response to Decision Letter 2]

6 Mar 2026

6. Review Comments to the Author

Reviewer #1: The revision of the manuscript is adequate and the authors have addressed most comments. A few remaining points:

1. Please correct grammatical errors and typos throughout. (e.g., line 87 promotes; line 100 awkward; line 190 that; line 542 might commanded, line 567 awkward clause at end of sentence; several more in discussion and other parts of the manuscript)

We have addressed these issues by correcting typos and rewording. As well we found a few other minor grammar issues and changed a bit of the wording throughout for clarity (shown in the “track changes” version).

2. Line 151: The RPE is mentioned here for the first time, without explanation or prior definition (abstract doesn’t count). Please fix.

We have defined RPE within the main text.

3. Figure captions for figures 3 and 4 say error bars are omitted, even though error bars are now included.

The figure captions have been modified accordingly.

4. Around line 534, the authors could acknowledge that the effect of compound type in Experiment 1b is small (in addition to ‘direct’) to cohere better with the results section (which says ‘slightly greater’), and lines 596-97 in the discussion, where potential reasons are given for the small effect size.

Though the point appears belabored, we have added that the effect is small in line 545.

---

## [Editor Report · Decision Letter 2]

13 Mar 2026

Perceptual learning mechanisms with single-stimulus exposure

PONE-D-25-32174R2

Dear Dr. SANJUAN ARTEGAIN,

We’re pleased to inform you that your manuscript has been judged scientifically suitable for publication and will be formally accepted for publication once it meets all outstanding technical requirements.

Kind regards,

Poppy Watson

Academic Editor

PLOS One
---

## [Editor Report · Acceptance letter]

PONE-D-25-32174R2

PLOS One

Dear Dr. SANJUAN ARTEGAIN,

I'm pleased to inform you that your manuscript has been deemed suitable for publication in PLOS One. Congratulations! Your manuscript is now being handed over to our production team.

Kind regards,

on behalf of

Dr. Poppy Watson

Academic Editor

PLOS One